

# Content quality and reliablity of YouTube™ videos as a source of information about good oral hygiene practices in adults

Gizem Ince Kuka and Hare Gursoy

Department of Periodontology, University of Health Sciences, Hamidiye Dental Faculty, Istanbul, Turkey

## ABSTRACT

**Background:** Good oral hygiene is crucial for preventing dental caries and periodontal diseases. However, proper and regular application of oral hygiene practices requires adequate knowledge. In recent years, the internet has become one of the most popular places to find health-related information, necessitating studies that analyze the quality of the content available online. The purpose of the present study was to analyze the content quality and reliability of YouTube™ videos on the topic of adult oral hygiene practices and to guide oral health care professionals who use this platform for patient education.

**Methods:** A YouTube™ search was performed of the most frequent search term, 'dental hygiene'. A total of 150 videos were screened, and 51 were included in the final study. The characteristics, sources, and content of the videos were analyzed using the Global Quality Score (GQS) and DISCERN reliability indices. The IBM SPSS 25 program was used for statistical analyses.

**Results:** Most of the included videos were uploaded by oral health care professionals (63%). GQS revealed only 17.6% of the videos were excellent quality whereas 23.5% of them were poor quality. In the content analysis, 62.7% of the videos were deemed moderately useful. Video duration, total content score, and interaction indices were all significantly higher in the useful and very useful groups compared to the slightly useful group ($p = 0.020$, $p < 0.001$, $p = 0.040$). GQS had a positive, low-medium statistically significant correlation with both video duration and total content scores ($r = 0.235$, $r = 0.517$; $p < 0.05$). DISCERN score also had a positive, low-medium statistically significant correlation with total content score ($r = 0.500$; $p < 0.05$).

**Conclusion:** The study concluded that most YouTube™ videos on oral hygiene practices for adults are moderately useful. When using YouTube™ for patient education, oral health care professionals and organizations should be aware of low-quality videos and seek out accurate, useful videos. There is also a need for quality videos with expanded oral health content.

# INTRODUCTION

Oral health is defined by the World Health Organization as the state in which individuals perform essential functions such as chewing and speaking without pain, discomfort, and

Corresponding author
Gizem Ince Kuka,
gizem.incekuka@sbu.edu.tr

embarrassment (*World Health Organization, 2023*). According to the Global Oral Health Status Report, untreated dental caries and severe periodontal diseases are among the leading causes of oral diseases, which are reported to affect approximately half of the global population (*Jain et al., 2023*). Oral health can be mostly maintained with proper oral hygiene practices and regular professional prophylaxis (*Lang & Bartold, 2018*). When oral hygiene is neglected, dental biofilm matures to its maximum extent within a 4-day period, initiating the clinical signs of gingival inflammation (*Petersilka, Ehmke & Flemmig, 2002*).

Oral health care professionals are primarily responsible for giving patients oral health-related information and oral hygiene instructions, which should be tailored to the needs of each individual. However, patients now tend to obtain their health-information from the internet, especially those who have difficulty accessing healthcare quickly (*Medlock et al., 2015*; *Hesse et al., 2005*). One survey found that 48.6% of adults in the US prefer to get first-stage health information online instead of visiting a physician (*Hesse et al., 2005*). Although online information is both free and easy to access, the quality of online sources is debatable (*Chu et al., 2017*). Thus, the widespread availability of uncontrolled information on digital platforms poses a potential risk to public health.

YouTube™ is a US-based social media platform that was launched in 2005 and has become a popular information database for a variety of subjects (*Yalcin-Ulker & Duygu, 2023*). The public has free access to this platform, increasing the distribution of knowledge (*Aksoy & Topsakal, 2022*). However, this access can also be harmful for patients because of possible misleading information disseminated from unreliable sources (*Duman, 2020*). Numerous YouTube™ videos are available about different medical conditions (*de Oliveira Júnior et al., 2023*; *Desai et al., 2013*). A review of related literature on dentistry revealed several publications that evaluate the quality of online videos on pediatric oral care (*Aksoy & Topsakal, 2022*; *Duman, 2020*; *Erturk-Avunduk & Delikan, 2023*), peri-implantitis (*Göller Bulut, Paksoy & Ustaoğlu, 2023*), oral hygiene practices during the COVID-19 pandemic (*Oz & Kirzioglu, 2021*), orthodontic treatment (*Ozturk & Gumus, 2022*), implants (*Abukaraky et al., 2018*), denture care (*Yagci, 2023*) and tooth extractions. However, the present study is the first to evaluate the quality of YouTube™ videos on adult oral hygiene practices. Unreliable and misleading information on this topic may harm patients, as improper brushing and interdental cleaning techniques, as well as the use of inaccurate devices, are associated with gum recession and a lack of professional dental care. This study hypothesized that the quality and content of YouTube™ videos about oral hygiene practices in adults would be generally insufficient and that the content of videos uploaded by professionals would be better than those uploaded by laypeople.

## MATERIALS AND METHODS

The present cross-sectional study did not require ethical approval since it contained open-access data. The Google Trends website (https://trends.google.com), which is an internet search engine designed to find the frequency of searches for particular terms over a given time frame, was used to obtain the most preferred search term within the 'last 5 years worldwide'. Possible keywords were searched, and 'dental hygiene' was identified as the most frequent search term related to oral hygiene practices (Fig. 1).
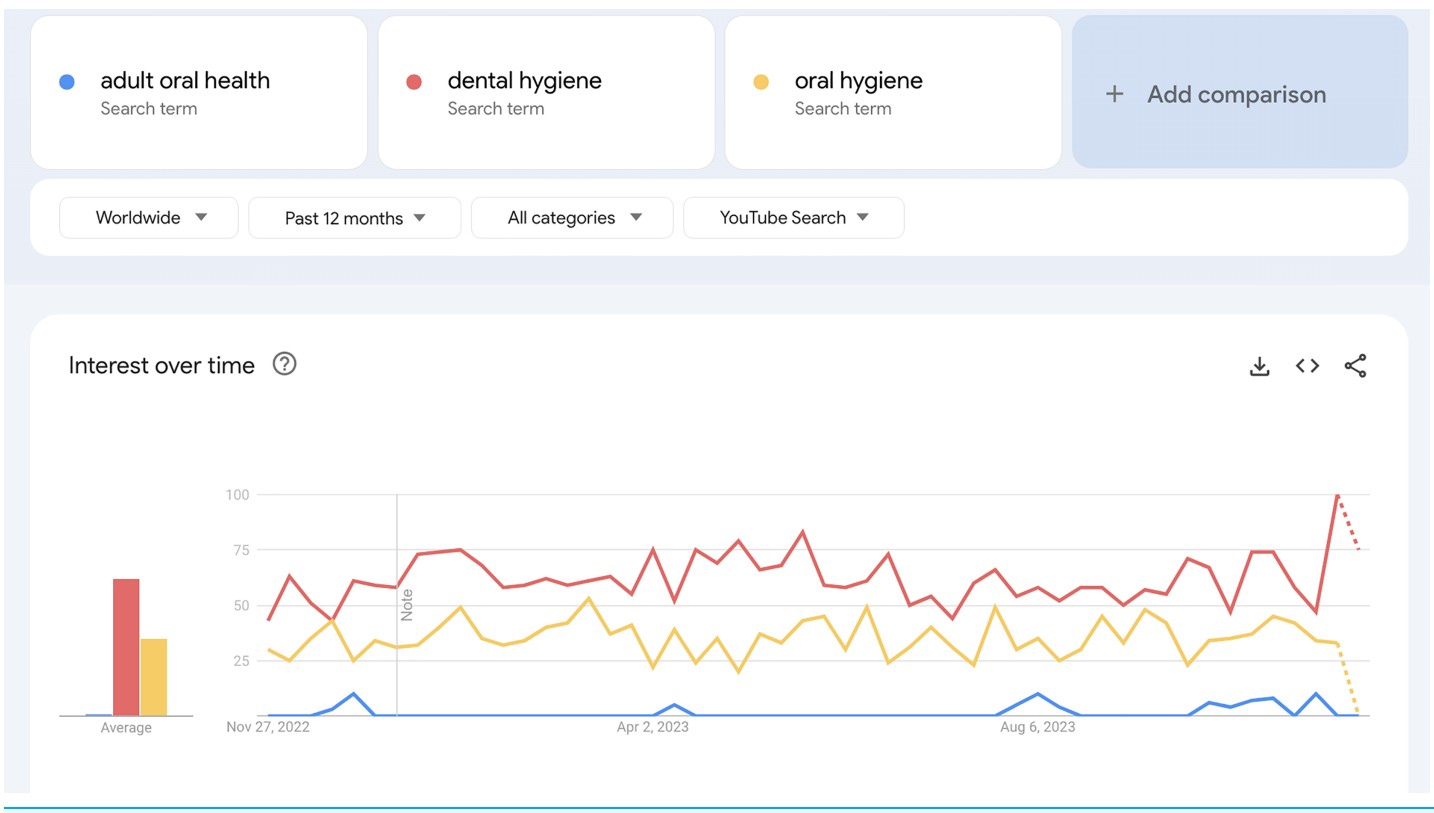

**Figure 1** Distribution of the search terms in Google Trends.

On September 18, 2023, a YouTube™ search was performed using the key word 'dental hygiene' without applying any filters. An experienced periodontist performed the search and assessed the videos between September 18 and 25. A total of 150 videos were screened for eligibility, and uniform resource locators (URLs) were recorded. The study included videos that provided oral health instructions in English, and excluded the following: (1) irrelevant videos; (2) commercials; (3) duplicates; (4) dental hygiene instructions for children, orthodontic patients, or denture care.

Descriptive data for the remaining videos was recorded, including the upload source, video duration, and number of views, likes, and comments. Video sources were categorized into the following: (1) oral health care professionals (dental hygienists, dentists, and dental specialists); (2) universities, clinics, and professional organizations; (3) online information websites (online dental platforms, TV channels); (4) laypeople. The interaction index percentage was calculated using the method described by *Duman (2020)*: (number of likes – dislikes/number of views) × 100. A content analysis (*Oz & Kirzioglu, 2021*), DISCERN (*Charnock et al., 1999*; *Singh, Singh & Singh, 2012*), and Global Quality Scores (GQS) (*Bernard et al., 2007*) were applied to the final 51 videos.

The 14 parameters shown in Fig. 2 and outlined by *Oz & Kirzioglu (2021)* guided the content analysis of the included videos. The questions in the analysis mainly addressed oral hygiene instructions and items related to healthy nutritional habits. Each criteria was scored as 0 (no) or 1 (yes), with a maximum possible score of 14 points. Video content was

| Scored Parameters<br>Yes: Score 1<br>No: Score 0 |
|---|
| Brushing teeth with a toothpaste twice daily, for at least 2 minutes<br>Tooth brushing techniques<br>Change of toothbrush after 3 months<br>Change of toothbrush after been cold<br>Never share toothbrush<br>Cleaning and disinfection of the toothbrush after each use<br>Avoid placing tooth brushes of a family in a common toothbrush holder<br>Tongue cleaning once daily<br>Interproximal cleaning everyday with interdental brushes or floss<br>Demonstration videos<br>Use of antiseptic mouthwashes<br>Staying hydrated<br>Limitation of the consumption of sweets, acidic foods, sugary beverages and snacking<br>Suggestion of the healthy nutrition such as consumption of vegetables, fruits, and nuts |

**Figure 2 Content quality scores: 0 = not useful; 1–3 = slightly useful; 4–7 = moderately useful; 8–11 = useful; 12–14 = very useful.**

| DISCERN Reliabilty Score (Yes: Score 1, No: Score 0) |
|---|
| 1. Are the aims clear and achieved? |
| 2. Are reliable sources of information used? |
| 3. Is the information presented balanced and unbiased? |
| 4. Are additional sources of information listed for patient reference? |
| 5. Are areas of uncertainty mentioned? |

| Global Quality Score (GQS) |
|---|
| 1. Poor quality, poor flow of video, most information missing, not at all useful for patients |
| 2. Generally poor quality and poor flow, some information listed but many important topics missing, of very limited use to patients |
| 3. Moderate quality, suboptimal flow, some important information is adequately discussed but others poorly discussed, somewhat useful for patients |
| 4. Good quality and generally flow. Most of the relevant information is listed, but some topics not covered, useful for patients. |
| 5. Excellent quality and flow, very useful for patients |

**Figure 3 DISCERN and Global Quality Score (GQS) parameters.**

classified according to the total score for each video: 0 points = not useful; 1–3 points = slightly useful; 4–7 points = moderately useful; 8–11 points = useful; and 12–14 points = very useful.

The DISCERN scale was used to score the reliability of health-related information. In addition to evaluating video content, this scale questions the source and objectivity of information and whether there is bias or conflict of interest. It includes five questions that can be either scored 0 (no) or 1 (yes) for a maximum of five points, with higher scores indicating more reliable content (Fig. 3).

GQS, which is a 5-point scale system, was used to evaluate video quality. This scoring system evaluates the researcher's general opinion, with video content quality ranging from poor (1) to excellent (5; Fig. 3).

## Statistical analysis

Descriptive statistics are given as the number, percentage, mean, standard deviation, minimum, maximum, and median. Analyses were carried out in the IBM SPSS 25 program. The distribution of normality was checked using the Shapiro-Wilk test. The Kruskal Wallis test was used to compare the means of three or more groups without a normal distribution. A *post-hoc* Bonferroni test was applied to reveal the group or groups that made the difference. Fisher's Exact test was applied in cases where the sample size assumption (expected value > 5) was not met in testing the relationship between categorical variables. Associations between an ordered categorical variable and continuous variables were checked with Kendall's Tau correlation. A $p$ value < 0.05 was considered statistically significant.

# RESULTS

## Descriptive variables

The first 150 videos in the search results that were related to the topic of adult oral hygiene were screened, and 51 of them were included in the present study (34%). Most of the included videos (62.7%) were uploaded by oral health care professionals, followed by universities, clinics, and professional organizations (25.5%), laypeople (7.8%), and online information websites (3.9%). Table 1 displays the descriptive characteristics of these videos, including duration, likes, views, comments, and interaction index scores. The included videos had an average overall content score of 6.86. The mean GQS and DISCERN scores were 3.53 and 2.96, respectively.

## GQS, content analysis, and DISCERN scores

Table 2 reveals the distribution of the videos in each of the evaluated indices. GQS revealed 41.2% of the videos ($n = 21$) were good quality, whereas 23.5% of them ($n = 12$) were generally poor quality. In the content analysis, 62.7% and 25.5% of the videos ($n = 32$, $n = 13$) were found to be moderately useful and useful, respectively. The DISCERN scores were distributed as follows: 29.4% ($n = 15$) received a score of 3, 25.5% ($n = 13$) scored a 4, and 23.5% ($n = 12$) scored a 2.

Comparison of GQS groups revealed statistically significant differences between the number of comments and total content scores ($p = 0.044$, $p < 0.001$; Table 3). According to the *post hoc* test results performed for double comparisons, there were more comments on videos of moderate quality than on videos of generally poor quality ($p = 0.027$). For total content scores, significant differences were detected between generally poor-quality videos and those of good quality and excellent quality ($p = 0.005$ and $p = 0.001$). Total content scores of good and excellent-quality videos were higher than those of generally poor-quality videos.
**Table 1 Descriptive characteristics of the included videos.**

|  | Minimum | Maximum | Mean | SD | Median |
|---|---|---|---|---|---|
| Duration | 0.92 | 29.93 | 8.20 | 6.80 | 6.58 |
| Views | 56.00 | 33,000,000.00 | 1,142,723.90 | 4,690,614.19 | 116,000.00 |
| Likes | 0.00 | 92,000.00 | 9,125.22 | 17,950.39 | 1,700.00 |
| Comments | 0.00 | 5,800.00 | 438.38 | 949.80 | 114.00 |
| Total content score | 3.00 | 13.00 | 6.86 | 2.39 | 7.00 |
| DISCERN | 0.00 | 5.00 | 2.96 | 1.28 | 3.00 |
| GQS | 2.00 | 5.00 | 3.53 | 1.05 | 4.00 |
| Interaction index | 0.00 | 514.29 | 12.42 | 71.84 | 1.64 |

Note:
GQS, Global quality scale; SD, Standard deviation.

**Table 2 Distribution of the videos according to the evaluated indices.**

|  |  | n | % |
|---|---|---|---|
| GQS | Poor | 0 | 0.0 |
|  | Generally poor | 12 | 23.5 |
|  | Moderate | 9 | 17.6 |
|  | Good | 21 | 41.2 |
|  | Excellent | 9 | 17.6 |
| DISCERN score | 0 | 2 | 3.9 |
|  | 1 | 3 | 5.9 |
|  | 2 | 12 | 23.5 |
|  | 3 | 15 | 29.4 |
|  | 4 | 13 | 25.5 |
|  | 5 | 6 | 11.8 |
| Total content score | Slightly useful | 4 | 7.8 |
|  | Moderately useful | 32 | 62.7 |
|  | Useful | 13 | 25.5 |
|  | Very useful | 2 | 3.9 |

Note:
GQS, Global quality scale.

**Table 3 Comparison of the video characteristics according to the GQS scores.**

|  |  | Median (minimum–maximum) | Mean ± S.D. | Test statistics | p |
|---|---|---|---|---|---|
| Duration | Generally poor quality | 2.79 (0.92–11.23) | 5.01 ± 4.03 | 5.423 | 0.143 |
|  | Moderate quality | 8.32 (1.93–11.52) | 6.81 ± 4.54 |  |  |
|  | Good quality | 8.42 (1.2–29.17) | 9.86 ± 7.74 |  |  |
|  | Excellent quality | 9.28 (3.6–29.93) | 9.97 ± 8.26 |  |  |
| Views | Generally poor quality | 67,500 (56–539,000) | 136,179.67 ± 182,667.95 | 3.210 | 0.360 |
|  | Moderate quality | 306,000 (3,500–5,900,000) | 1,330,055.56 ± 2,170,135.4 |  |  |
|  | Good quality | 145,000 (186–4,154,358) | 510,338.62 ± 944,091.5 |  |  |
|  | Excellent quality | 67,000 (7,300–33,000,000) | 3,773,016.89 ± 10,960,648 |  |  |

|  |  | Median (minimum–maximum) | Mean ± S.D. | Test statistics | p |
|---|---|---|---|---|---|
| Likes | Generally poor quality | 525 (0–17,000) | 3,176.92 ± 5,617.08 | 7.445 | 0.059 |
|  | Moderate quality | 6,500 (671–92,000) | 20,919 ± 30,842.81 |  |  |
|  | Good quality | 1,900 (3–68,000) | 10,047.29 ± 17,364.75 |  |  |
|  | Excellent quality | 1,700 (38–9,800) | 3,111 ± 3,588.34 |  |  |
| Comments | Generally poor quality | 39.5 (0–605) | 113.42 ± 183.6 | 8.113 | **0.044** |
|  | Moderate quality | 347 (93–3,029) | 878.86 ± 1,050.11 |  |  |
|  | Good quality | 135 (0–5,800) | 574.25 ± 1,283.06 |  |  |
|  | Excellent quality | 124 (4–950) | 227.11 ± 303.61 |  |  |
| Total content score | Generally poor quality | 5 (3–7) | 4.75 ± 1.36 | 17.743 | **<0.001** |
|  | Moderate quality | 6 (5–10) | 6.44 ± 1.59 |  |  |
|  | Good quality | 7 (5–11) | 7.29 ± 1.74 |  |  |
|  | Excellent quality | 10 (3–13) | 9.11 ± 3.18 |  |  |
| Interaction index | Generally poor quality | 1.22 (0–4.36) | 1.68 ± 1.63 | 1.941 | 0.585 |
|  | Moderate quality | 2.12 (0.8–514.29) | 58.74 ± 170.83 |  |  |
|  | Good quality | 1.64 (0.22–34.68) | 3.41 ± 7.26 |  |  |
|  | Excellent quality | 1.39 (0.03–3.33) | 1.43 ± 1.15 |  |  |

**Note:**
Kruskal Wallis test, $p < 0.05$. Bold values indicate statistical significance.

**Table 4 Comparison of the video characteristics according to the total content score.**

|  |  | Median (minimum–maximum) | Mean ± S.D. | Test statistics | p |
|---|---|---|---|---|---|
| Duration | Slightly useful | 3.41 (0.92–4.88) | 3.15 ± 2.02 | 7.786 | **0.020** |
|  | Moderately useful | 5.8 (1.2–16.18) | 6.8 ± 4.57 |  |  |
|  | Useful & very useful | 9.93 (3.6–29.93) | 12.53 ± 9.28 |  |  |
| Views | Slightly useful | 14,050 (56–33,000,000) | 8,257,039 ± 16,495,312.04 | 1.531 | 0.465 |
|  | Moderately useful | 142,500 (186–5,900,000) | 564,597.81 ± 1,241,975.76 |  |  |
|  | Useful & very useful | 67,000 (975–4,154,358) | 478,908.87 ± 1,076,717.04 |  |  |
| Likes | Slightly useful | 26.5 (0–9,800) | 2,463.25 ± 4,891.18 | 3.351 | 0.187 |
|  | Moderately useful | 1,800 (3–92,000) | 9,138.03 ± 18,440.68 |  |  |
|  | Useful & very useful | 1,700 (27–68,000) | 10,874.4 ± 19,408.77 |  |  |
| Comments | Slightly useful | 30 (0–950) | 252.5 ± 465.7 | 1.261 | 0.532 |
|  | Moderately useful | 104 (0–3,029) | 363.07 ± 630.71 |  |  |
|  | Useful & very useful | 242 (4–5,800) | 633.53 ± 1,457.41 |  |  |
| Interaction index | Slightly useful | 0.07 (0–1.91) | 0.51 ± 0.93 | 6.450 | **0.040** |
|  | Moderately useful | 1.63 (0.1–514.29) | 17.82 ± 90.6 |  |  |
|  | Useful & very useful | 2.13 (0.29–34.68) | 4.08 ± 8.51 |  |  |

**Note:**
Kruskal Wallis test, $p < 0.05$. Bold values indicate statistical significance.

**Table 5 Comparison of the video characteristics according to the DISCERN score.**

| | | Median (minimum–maximum) | Mean ± S.D. | Test statistics | p |
|---|---|---|---|---|---|
| Duration | 0 and 1 | 11.23 (1.97–29.17) | 12.69 ± 10.74 | 8.578 | 0.073 |
| | 2 | 4.3 (0.92–11.52) | 5.56 ± 4.02 | | |
| | 3 | 6.89 (1.93–13.32) | 7.01 ± 4.58 | | |
| | 4 | 6.48 (1.2–15.5) | 6.76 ± 3.82 | | |
| | 5 | 14.13 (3.6–29.93) | 16.55 ± 10.51 | | |
| Views | 0 and 1 | 39,000 (3,400–33,000,000) | 6,893,880 ± 14,605,943.6 | 2.593 | 0.628 |
| | 2 | 98,000 (56–879,000) | 194,800.53 ± 248,883.44 | | |
| | 3 | 179,500 (975–5,900,000) | 1.299,458.08 ± 2,131,023.62 | | |
| | 4 | 199,000 (528–1,558,000) | 366,732.92 ± 431,385.93 | | |
| | 5 | 54,000 (186–239,000) | 87,747.67 ± 96,343.56 | | |
| Likes | 0 and 1 | 1,700 (32–18,000) | 5,913.6 ± 7,875.26 | 5.576 | 0.233 |
| | 2 | 671 (0–17,000) | 3,690.8 ± 5,758.6 | | |
| | 3 | 5,100 (27–92,000) | 20,721.33 ± 31,422.03 | | |
| | 4 | 3,900 (3–43,000) | 9,266.77 ± 13,157.51 | | |
| | 5 | 402 (10–6,700) | 1,888.67 ± 2,741.73 | | |
| Comments | 0 and 1 | 67 (0–951) | 394 ± 508.73 | 5.709 | 0.222 |
| | 2 | 48 (0–668) | 156.43 ± 223.35 | | |
| | 3 | 300 (7–5,800) | 1,159.4 ± 1,881.47 | | |
| | 4 | 210 (0–1,214) | 337.77 ± 366.27 | | |
| | 5 | 22.5 (0–606) | 149.5 ± 242.05 | | |
| Total content score | 0 and 1 | 5 (3–9) | 5.4 ± 2.61 | 19.851 | **0.001** |
| | 2 | 5 (3–7) | 5.27 ± 1.33 | | |
| | 3 | 7 (5–10) | 7 ± 1.71 | | |
| | 4 | 7 (5–12) | 7.62 ± 2.26 | | |
| | 5 | 10 (7–13) | 10.17 ± 1.94 | | |
| Interaction index | 0 and 1 | 1.06 (0.03–4.36) | 1.37 ± 1.76 | 3.894 | 0.421 |
| | 2 | 1.18 (0–4.09) | 1.5 ± 1.24 | | |
| | 3 | 2.16 (0.29–514.29) | 44.66 ± 147.9 | | |
| | 4 | 1.97 (0.22–34.68) | 4.2 ± 9.2 | | |
| | 5 | 1.88 (0.38–5.38) | 2.25 ± 1.74 | | |

**Note:**
Kruskal Wallis test, $p < 0.05$. Bold values indicate statistical significance.

In the total content score categories, significant differences were found between video duration and interaction index scores ($p = 0.02$, $p = 0.04$; Table 4). According to the *post hoc* tests performed for video duration, significant differences were detected between the useful group and the moderately useful and slightly useful groups ($p < 0.001$ and $p < 0.001$). Videos with a useful score had longer durations than videos with moderately or slightly useful scores. The interaction index scores of useful videos were higher than those of the slightly useful group ($p = 0.034$).

Significant differences were detected in the comparison of DISCERN score groups and total content scores ($p = 0.001$; Table 5). The *post hoc* tests determined that videos with a

**Table 6 Comparison of GQS, DISCERN, and total content scores.**

|  |  | Generally poor quality | | Moderate quality | | Good quality | | Excellent quality | | | |
|---|---|---|---|---|---|---|---|---|---|---|---|
|  |  | n | % | n | % | n | % | n | % | Test statistics | p |
| DISCERN | 0 and 1 | 2 | 16.7 | 0 | 0.0 | 2 | 9.5 | 1 | 11.1 | 39.002 | **<0.001** |
|  | 2 | 10 | 83.3 | 3 | 33.3 | 1 | 4.8 | 1 | 11.1 |  |  |
|  | 3 | 0 | 0.0 | 6 | 66.7 | 5 | 23.8 | 1 | 11.1 |  |  |
|  | 4 | 0 | 0.0 | 0 | 0.0 | 10 | 47.6 | 3 | 33.3 |  |  |
|  | 5 | 0 | 0.0 | 0 | 0.0 | 3 | 14.3 | 3 | 33.3 |  |  |
| Total content score | Slightly useful | 3 | 25.0 | 0 | 0.0 | 0 | 0.0 | 1 | 11.1 | 22.181 | **<0.001** |
|  | Moderately useful | 9 | 75.0 | 8 | 88.9 | 14 | 66.7 | 1 | 11.1 |  |  |
|  | Useful & very useful | 0 | 0.0 | 1 | 11.1 | 7 | 33.3 | 7 | 77.8 |  |  |

Note:
Fisher's Exact test, $p < 0.05$. Bold values indicate statistical significance.

**Table 7 Correlation between video characteristics and GQS, DISCERN, and total content scores.**

|  |  | GQS | DISCERN | Total content score |
|---|---|---|---|---|
| Duration | r | 0.235 | 0.137 | 0.319 |
|  | p | **0.029** | 0.194 | **0.005** |
| Views | r | 0.072 | −0.002 | 0.039 |
|  | p | 0.505 | 0.987 | 0.731 |
| Likes | r | 0.050 | 0.039 | 0.128 |
|  | p | 0.644 | 0.712 | 0.259 |
| Comments | r | 0.069 | 0.032 | 0.122 |
|  | p | 0.536 | 0.769 | 0.297 |
| Interaction index | r | −0.019 | 0.151 | 0.228 |
|  | p | 0.857 | 0.152 | **0.045** |

Note:
Kendal's Tau correlation, $p < 0.05$. Bold values indicate statistical significance.

DISCERN score of 5 had significantly higher total content scores than those with scores of 0, 1, or 2 ($p = 0.025$).

### Correlations between GQS, content analysis, and DISCERN scores

There were statistically significant differences in the comparison of GQS groups with DISCERN and total content score groups ($p < 0.001$, $p < 0.001$; Table 6). The correlation analysis between video characteristics and GQS, DISCERN, and total content scores found GQS had a low-medium statistically significant correlation with both video duration and total content scores (correlation coefficients of 0.235 and 0.517; $p = 0.029$, $p < 0.001$; Table 7). There were no statistically significant differences between video sources and GQS, DISCERN, and total content scores ($p > 0.05$; Table 8).

### DISCUSSION

The results of the present study revealed statistically significant, positive associations between the GQS, total content score, and DISCERN indexes. Videos of good quality had

**Table 8 Comparison of the video sources and their Global Quality Scores, DISCERN and total content scores.**

| | | Oral health care professionals | | Online information website | | Universities, dental clinics, professional organization | | Laypeople | | | |
|---|---|---|---|---|---|---|---|---|---|---|---|
| | | n | % | n | % | n | % | n | % | Test statistics | p |
| GQS | Generally poor | 6 | 18.8 | 1 | 50.0 | 3 | 23.1 | 2 | 50.0 | 11.674 | 0.110 |
| | Moderate | 3 | 9.4 | 0 | 0.0 | 5 | 38.5 | 1 | 25.0 | | |
| | Good | 17 | 53.1 | 0 | 0.0 | 3 | 23.1 | 1 | 25.0 | | |
| | Excellent | 6 | 18.8 | 1 | 50.0 | 2 | 15.4 | 0 | 0.0 | | |
| DISCERN | 0 ve 1 | 3 | 9.4 | 1 | 50.0 | 1 | 7.7 | 0 | 0.0 | 12.236 | 0.293 |
| | 2 | 6 | 18.8 | 1 | 50.0 | 5 | 38.5 | 3 | 75.0 | | |
| | 3 | 7 | 21.9 | 0 | 0.0 | 4 | 30.8 | 1 | 25.0 | | |
| | 4 | 10 | 31.3 | 0 | 0.0 | 3 | 23.1 | 0 | 0.0 | | |
| | 5 | 6 | 18.8 | 0 | 0.0 | 0 | 0.0 | 0 | 0.0 | | |
| Total content | Slightly useful | 2 | 6.3 | 1 | 50.0 | 1 | 7.7 | 0 | 0.0 | 8.997 | 0.103 |
| | Moderately useful | 17 | 53.1 | 1 | 50.0 | 11 | 84.6 | 3 | 75.0 | | |
| | Useful & very useful | 13 | 40.6 | 0 | 0.0 | 1 | 7.7 | 1 | 25.0 | | |

**Note:**
Fisher's Exact test, $p < 0.05$.

more comments, and according to the total content scores, useful videos had higher interaction index scores and longer durations than slightly useful videos; however, no significant difference was detected in the number of views or likes. The majority of the related videos were moderately useful, without any difference according to the source of upload. Therefore, the hypotheses of the study were rejected. This study's findings are important in that they reveal the need for oral health care professionals and organizations to add videos with higher quality and expanded content regarding oral hygiene practices in adults. Most of the included videos provided information about toothbrushing twice a day with toothpaste (82%), interdental cleaning (80%), and demonstration videos (68%). These results are in line with the findings of *Oz & Kirzioglu (2021)*, who evaluated the quality of the videos on oral hygiene practices during the COVID-19 pandemic period. However, the topics of toothbrush renewal frequency (26%), toothbrush disinfection (20%), and tongue cleaning (43%) were rarely mentioned. Oral health care professionals and organizations should be aware of these results and focus on providing additional information on these oral hygiene topics that are lacking in current videos. Health care professionals should also look for appropriate videos when using YouTube[TM] for patient education.

The rise of technology has led to inevitable changes in human behavior, habits, and daily routines. The power of internet access makes lives easier by providing information faster than conventional search methods (*Divakar, 2023*). Due to the speed of internet answers, most people tend to obtain health-related information online. YouTube[TM] is an online platform where users can upload whatever content they want, and it is reported to be the second most-visited website worldwide (*Yalcin-Ulker & Duygu, 2023*). A recent study

reported that individuals use YouTube™ to not only gain knowledge but also to make decisions (*Mohamed & Shoufan, 2024*; *Smyth et al., 2020*). Although there are many healthcare-related YouTube™ videos available, the quality, reliability, and content of these videos can be questionable (*Aksoy & Topsakal, 2022*). Therefore, professional evaluation and criticism of these videos is of the utmost importance (*Yagci, 2023*). Misleading and poor-quality videos put YouTube™ users at risk of making wrong choices based on inaccurate information (*Mohamed & Shoufan, 2024*).

Previous studies have indicated that individuals who want to get information from YouTube™ screen the initial 60–200 videos of a search, but only view the first 30 of them (*Aksoy & Topsakal, 2022*; *Duman, 2020*; *Desai et al., 2013*). One study reported that most users view only the first three pages of a search, which includes the first 120 videos (*Aksoy & Topsakal, 2022*). Beacuse of this finding, the present study screened only the first 150 videos for eligilibity. A review of the literature presented a limited number of studies, with varying hypotheses and study designs, examining the quality and reliability of YouTube™ videos for patient education on oral hygiene practices in adult individuals (*Oz & Kirzioglu, 2021*; *Smyth et al., 2020*). Therefore, the findings of the present study may be considered pioneering in this field. In the present study, video duration and interaction indexes were significantly higher in useful videos compared to slightly useful ones, and good-quality videos had more comments lower-quality videos. The mean number of comments in this study was also higher than in previous studies assessing the content of oral hygiene YouTube™ videos during the COVID-19 pandemic (*Oz & Kirzioglu, 2021*) and in the pediatric population (*Aksoy & Topsakal, 2022*). The present study also found significant positive correlations between video duration, total content score, and GQS. These results suggest that higher-quality videos receive more reactions, as evidenced by an increase in comments and interaction index scores. This finding aligns with that of previous studies, which have also reported a positive relationship between video duration and quality (*Göller Bulut, Paksoy & Ustaoğlu, 2023*; *Oz & Kirzioglu, 2021*). High-quality videos may be longer because they cover more important content on oral hygiene practices. The findings of the present study were also in line with previous studies reporting a positive relationship between number of likes, comments, views, and total content scores, revealing viewer behavior as higher-quality content receives more attention (*Yalcin-Ulker & Duygu, 2023*; *Aksoy & Topsakal, 2022*; *Göller Bulut, Paksoy & Ustaoğlu, 2023*). However, there are also studies that failed to observe any relationship between video duration, interaction index, and quality (*Duman, 2020*; *Oz & Kirzioglu, 2021*). This finding in these studies was attributed to the viewer's loss of interest as the video duration increased, despite high-quality video content.

There are also some limitations of the present study. Due to the nature of the online video sharing platforms, the study results may differ if performed in a different time period since videos are constantly being removed and added. Additionally, the video search and analysis were performed by a single researcher, and the results reflect this researcher's evaluation. Only including videos in English also did not capture useful videos in different languages.

## CONCLUSIONS

Most of the adult oral hygiene YouTube[TM] videos were moderately useful, but lacked some important information, regardless of the upload source. Proper education of adults on oral hygiene is crucial for the prevention of caries and periodontal disease. Although oral health care professionals and organizations still have a pivotal role in disseminating information, the digital age has forced the modification of these conventional methods. In this context, the reliability and content quality of oral health-related videos should be checked regularly by oral health care professionals. New videos with enhanced content and different subtopics should be uploaded by universities or professional organizations in official channels specifically designed for sharing official healthcare-related information. Within the study limitations, it was concluded that further studies evaluating the video content of oral health-related information on different digital platforms are required.

### Funding

The authors received no funding for this work.

### Competing Interests

The authors declare that they have no competing interests.

### Author Contributions

- Gizem Ince Kuka conceived and designed the experiments, performed the experiments, prepared figures and/or tables, authored or reviewed drafts of the article, and approved the final draft.
- Hare Gursoy analyzed the data, prepared figures and/or tables, authored or reviewed drafts of the article, and approved the final draft.

### Data Availability

The raw data is available in the Supplemental File.

### Supplemental Information

Supplemental information for this article can be found online at http://dx.doi.org/10.7717/peerj.18183#supplemental-information.

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
