# Peer review of "Content quality and reliablity of YouTube™ videos as a source of information about good oral hygiene practices in adults"

_PeerJ, doi:10.7717/peerj.18183_

## Round 0.1 · original submission · Major Revisions

The authors are requested to carefully revise the manuscript and answer the questions raised by the reviewers.

Reviewer 1 ·

Basic reporting

no comment

Experimental design

no comment

Validity of the findings

no comment

Additional comments

no comment

Annotated reviews are not available for download in order to protect the identity of reviewers who chose to remain anonymous.

Reviewer 2 ·

Basic reporting

-Conduct a thorough proofreading pass to correct grammatical errors, such as subject-verb agreement, punctuation, and sentence fragments.
-Ensure the use of formal academic language appropriate for the subject matter and audience.
-Ensure consistent use of tense (past tense for methods and results, present tense for general statements) throughout the document.
-Maintain a consistent voice (active or passive) within sentences and paragraphs.
-Use precise language to convey specific meanings and avoid ambiguity.
TO ENHANCE THE CLARITY AND QUALITY OF YOUR MANUSCRIPT, WE RECOMMEND CONSIDERING PROFESSIONAL ENGLISH EDITING SERVICES.- Clarify Sentence Structures and Remove Redundant Information.
- Ensure consistency in terminology. For example, "oral health care professionals" and "oral health-related information" should be consistently used throughout.
- The flow between sentences can be improved for better coherence.
- Ensure smooth transitions between paragraphs to maintain the logical flow of ideas.

Experimental design

The manuscript "Assessment of YouTubeTM videos as a source of knowledge regarding oral hygiene practices in adults" has been submitted to PeerJ.

The purpose of this research was to analyze the content and reliability of the YouTube™ videos regarding oral hygiene measures in adults and to guide dental professionals who use this platform for patient education. The study's findings indicate that most of the videos are moderately useful. Moreover, the authors indicate that dental health care professionals and organizations should be aware of this situation when using this platform for patient education.

While the manuscript tackles an important issue, there are several concerns regarding the study.

Specific comments are noted below:

Title: Include the type of study. Reliability and Content of YouTube Videos….could be a more appropriate title. "Assessment" may be replaced with "Reliability and Content" to better reflect the analysis performed.

Abstract
- The background section only presents the objective. Please review.
- Briefly mention the percentages of high/low-quality videos and the positive correlation between video duration/quality and content.
- A p-value of 0.000 is not meaningful; please include the first available significant digit. Consider this recommendation throughout the manuscript.
- It seems there are two groups: highly useful and slightly useful, but nothing is mentioned about this in the methodology.
- Please adjust the specific percentages mentioned based on the actual data for better accuracy.
- Instead of using the word "relationships" in statistical terms, it is better to use "associations" or "correlations."
- In line 34, are the authors referring to correlations? If so, please present the correlation value.
- It is recommended to properly adhere to appropriate statistical terms to understand the information presented in the results and conclusions. Moreover, Consider this recommendation throughout the manuscript.
- Emphasize the focus on dental professionals using YouTube for patient education.

Keywords: Please provide them and ensure that all of them correspond to MeSH terms.

Introduction
- Clearly state the problem being addressed by the study. This could be something like: "The widespread availability of unvetted oral hygiene information on YouTube poses a potential risk to public health."
- Frame the hypothesis as a concern based on the potential for unreliable information.
- Consider streamlining some information. For example, the sentence about YouTube being the second most-visited website (sentence 63) is not directly relevant to the problem and could be omitted.
- The introduction lacks emphasis on the potential negative consequences of unreliable oral hygiene information on YouTube. It doesn't delve into the specific risks associated with misleading information, such as improper brushing techniques leading to gum disease or delaying necessary dental care.
- The hypothesis (poor quality, low usefulness) could be more specific about the content and its potential impact.
- Clarify Sentence Structures and Remove Redundant Information.
- Ensure consistency in terminology. For example, "oral health care professionals" and "oral health-related information" should be consistently used throughout.
- The flow between sentences can be improved for better coherence.
- Ensure smooth transitions between paragraphs to maintain the logical flow of ideas.

M&M
- Improve Clarity and Grammar. The methodology section contains several grammatical errors and unclear sentences. Ensure consistent use of terminology and correct any misspellings.
- Clearly state the inclusion and exclusion criteria to avoid ambiguity.
- Ensure that data recording and analysis methods are clearly stated.
- Clearly explain the scoring systems and index calculations.
- Provide a detailed explanation of the content analysis and scoring systems used.
- Consider explaining why Google Trends was used and how it strengthens the study.
- More details on YouTube search could be helpful. For example, were there any language restrictions or filters applied besides relevance?
- Providing more information about the specific criteria used for content analysis (Figure 2) would enhance transparency.
- Briefly explain the specific topics covered in the DISCERN scale (mentioned in sentence 107).

Validity of the findings

Results
- Ensure each paragraph clearly presents a specific aspect of the results and use headings or subheadings if necessary to organize the content.
- Consistently use terms like "videos" instead of alternating between "videos" and "contents" to avoid confusion.
- Ensure that results are presented in a logical order, starting with descriptive statistics, followed by more detailed analyses and specific findings.
- Clearly explain what the statistical tests are showing and avoid jargon that may not be understood by all readers.

Discussion
- Briefly summarize the main results at the beginning.
- Expand on the impact of technology on health information dissemination and the role of YouTubeTM specifically. Provide a broader context on how online platforms influence health-related behaviors and decision-making processes.
- Emphasize more explicitly how this study contributes to the existing literature on YouTubeTM videos related to oral hygiene. Discuss why evaluating these videos is important and how this study fills a gap in current research.
- Clearly interpret the findings in relation to the study hypotheses. Discuss why the hypotheses were rejected and what implications this has for future research or clinical practice.
- Compare your findings with previous research on YouTubeTM video quality, reliability, and usefulness. Highlight similarities and differences, particularly in methodologies and outcomes.
- Based on the findings, provide specific recommendations for dental professionals, educators, and policymakers regarding the creation and dissemination of oral health information on YouTubeTM. Consider practical steps to improve the quality and accessibility of educational videos.
- This study has more limitations that were not recognized.

Conclusions
- Ensure that the conclusions succinctly summarize the main findings of the study, particularly emphasizing the level of usefulness of the YouTubeTM videos evaluated.
- Connect the conclusions back to the study objectives outlined in the introduction. - - Highlight how the findings address the research questions and hypotheses.
- Expand on the implications of the study findings for public health, emphasizing the importance of effective oral hygiene education in preventing common dental issues like caries and periodontal diseases.
- Provide actionable recommendations for dental professionals, educators, and policymakers based on the study findings. Include suggestions for improving the quality and reliability of online health-related content.
- Acknowledge the study's limitations and suggest avenues for future research.
-Conclude with a strong statement that reinforces the need for ongoing evaluation and improvement of online health information, particularly in the context of rapidly evolving digital platforms.

---

## Round 0.2 · accepted · Accept

After revisions, both reviewers agreed to publish the manuscript. I also reviewed the manuscript and found no obvious risks to publication. Therefore, I also approve the publication of this manuscript.

Reviewer 1 ·

Basic reporting

appropriate

Experimental design

appropriate

Validity of the findings

appropriate

Additional comments

none

Reviewer 2 ·

Basic reporting

The authors have made all requested suggestions/corrections.

Experimental design

The authors have made all requested suggestions/corrections.

Validity of the findings

The authors have made all requested suggestions/corrections.

Additional comments

The authors have made all requested suggestions/corrections.